# Cardiopulmonary Long-Term Sequelae in Patients after Severe COVID-19 Disease

**DOI:** 10.3390/jcm12041536

**Published:** 2023-02-15

**Authors:** Julia Hanne Niebauer, Christina Binder-Rodriguez, Ahmet Iscel, Sarah Schedl, Christophe Capelle, Michael Kahr, Simona Cadjo, Simon Schamilow, Roza Badr-Eslam, Michael Lichtenauer, Aurel Toma, Alexander Zoufaly, Rosmarie Valenta, Sabine Hoffmann, Silvia Charwat-Resl, Christian Krestan, Wolfgang Hitzl, Christoph Wenisch, Diana Bonderman

**Affiliations:** 1Department of Cardiology, Favoriten Clinic, 1100 Vienna, Austria; 2Department of Cardiology, Medical University of Vienna, 1090 Vienna, Austria; 3Department of Internal Medicine II, Division of Cardiology, University Hospital Salzburg, 5020 Salzburg, Austria; 4Department of Infectious Diseases, Favoriten Clinic, 1100 Vienna, Austria; 5Faculty of Medicine, Sigmund Freud University, 1020 Vienna, Austria; 6Department of Radiology, Favoriten Clinic, 1100 Vienna, Austria; 7Team Biostatistics and Publication of Clinical Trial Studies, Research and Innovation Management (RIM), Paracelsus Medical University, 5020 Salzburg, Austria; 8Department of Ophthalmology and Optometry, Paracelsus Medical University Salzburg, 5020 Salzburg, Austria; 9Research Program Experimental Ophthalmology and Glaucoma Research, Paracelsus Medical University, 5020 Salzburg, Austria

**Keywords:** SARS-CoV-2, pandemic, Long-COVID, fatigue, long-term impairment, heart, lung, myocarditis, risk factors

## Abstract

We aimed to identify cardiopulmonary long-term effects after severe COVID-19 disease as well as predictors of Long-COVID in a prospective registry. A total of 150 consecutive, hospitalized patients (February 2020 and April 2021) were included six months post hospital discharge for a clinical follow-up. Among them, 49% experienced fatigue, 38% exertional dyspnea and 75% fulfilled criteria for Long-COVID. Echocardiography detected reduced global longitudinal strain (GLS) in 11% and diastolic dysfunction in 4%. Magnetic resonance imaging revealed traces of pericardial effusion in 18% and signs of former pericarditis or myocarditis in 4%. Pulmonary function was impaired in 11%. Chest computed tomography identified post-infectious residues in 22%. Whereas fatigue did not correlate with cardiopulmonary abnormalities, exertional dyspnea was associated with impaired pulmonary function (OR 3.6 [95% CI: 1.2–11], *p* = 0.026), reduced GLS (OR 5.2 [95% CI: 1.6–16.7], *p* = 0.003) and/or left ventricular diastolic dysfunction (OR 4.2 [95% CI: 1.03–17], *p* = 0.04). Predictors of Long-COVID included length of in-hospital stay (OR: 1.15 [95% CI: 1.05–1.26], *p* = 0.004), admission to intensive care unit (OR cannot be computed, *p* = 0.001) and higher NT-proBNP (OR: 1.5 [95% CI: 1.05–2.14], *p* = 0.026). Even 6 months after discharge, a majority fulfilled criteria for Long-COVID. While no associations between fatigue and cardiopulmonary abnormalities were found, exertional dyspnea correlated with impaired pulmonary function, reduced GLS and/or diastolic dysfunction.

## 1. Introduction

The severe acute respiratory syndrome coronavirus type 2 (SARS-CoV-2) pandemic has had an influence on human lives worldwide, impacting global health and putting pressure on health care systems. Despite vaccines being introduced, many adverse events occurred [1].

Although the vast majority of patients return to their former state of health after acute COVID-19 disease, a considerable number continue to suffer from ongoing symptoms [2]. COVID-19 could have detrimental sequelae even after the post-acute phase, representing a new pathological condition: the “post-COVID-19 syndrome (PCS)” or “long-term COVID-19” [3]. Indeed, its prevalence has been estimated as high as 10–25% in ambulatory patients and becomes significantly higher when the infection has led to hospital admission [4]. Previous studies have shown that even 6 months after hospitalization for COVID-19 disease, as many as 68% of affected individuals still suffer from at least one symptom, such as fatigue or dyspnea, and 49% continue to do so even after 12 months [5]. Reduced performance on the 6 min walking distance test (6-MWT) as well as impaired pulmonary diffusion capacity 3 months after COVID-19 disease has been reported [6]. Although several studies have observed cardiopulmonary damage secondary to COVID-19 disease [7,8], neither cardiac magnetic resonance imaging (MRI) nor pulmonary function or cardiopulmonary exercise testing revealed pathologies explaining persistent symptom burden [5,6,7,8].

Although a broad spectrum of pathophysiological mechanisms have been put forward as potential causes of Long-COVID, such as direct virus-induced organ damage, persisting inflammation, immune dysregulation and/or hypercoagulability [9,10], the exact mechanism remains largely unknown. However, current evidence suggests that a more severe disease course is associated with a higher prevalence of Long-COVID [11,12].

Even though several studies assessed COVID-19 long-term effects, the percentage of patients suffering from long-term cardiopulmonary impairment remains unknown.

Therefore, it was the aim of this study to systematically screen previously hospitalized COVID-19 patients for remaining cardiopulmonary pathologies and to investigate potential associations with clinical symptoms attributed to Long-COVID.

## 2. Materials and Methods

### 2.1. Study Design and Setting

The present study was a prospective registry, which enrolled consecutive patients after PCR-confirmed COVID-19 disease and hospital treatment between February 2020 and April 2021. All study visits were performed at the Division of Cardiology, Favoriten Clinic, or at the Division of Cardiology, Medical University of Vienna.

Cardiac manifestation was assessed by echocardiography and defined as a reduced left ventricular function, reduced global longitudinal strain and/or diastolic dysfunction; postinfectious cardiac manifestations were documented by MRI. Pulmonary manifestation was defined as reduced pulmonary function and/or post-COVID-19 abnormality on the chest CT scan. In keeping with current recommendations, Long-COVID was defined as at least one persisting symptom 6 months after discharge, which had to be related to the acute infection independent from underlying and non-COVID-associated organ dysfunction [13,14].

The study itself was in line with the Declaration of Helsinki, as well as local laws, and was approved by the local ethics committee (EK 20-153-0720). All study participants gave their written informed consent.

### 2.2. Population

Patients had either been hospitalized on a regular ward or an intensive care unit (ICU) of our dedicated COVID-19 Division.

Consecutive patients were contacted by telephone calls without any preselection and invited to participate in the study 6 months post discharge. Exclusion criteria were age younger than 18 years as well as pregnancy.

### 2.3. Outcome Baseline Examinations

All participants were explored in a standardized manner and underwent a cardiac and pulmonary workup including a clinical assessment, six-minute walk test (6-MWT), an analysis of standard laboratory parameters, transthoracic echocardiography (TTE), cardiac MRI, pulmonary function test and chest CT scan in an ambulatory setting.

### 2.4. Clinical Assessment

Profound clinical examination with a 12-lead ECG (Amedtec ECG pro, Aue-Bad Schlema, Sachsen, Germany) and blood pressure and oxygen saturation measurements were performed in each patient. A detailed patient history was obtained, and a standardized list of the following symptoms was asked: exertional dyspnea, fatigue, fever, cough, myalgia, vertigo, loss/decrease of taste/smell, hair loss, neurological abnormalities and others.

### 2.5. Six-Minute Walk Test

The 6-MWT was performed on an indoor track to assess submaximal exercise capacity according to the American Thoracic Society guidelines [15].

### 2.6. Laboratory Parameters

A comprehensive list of laboratory parameters including cardiac biomarkers, such as N-terminal brain natriuretic peptide (NT-proBNP) and troponin T, was analyzed at the Department of Laboratory Medicine, Favoriten Clinic, Vienna, Austria.

### 2.7. Transthoracic Echocardiography

All TTE studies were performed by physicians in the echo laboratory of the Division of Cardiology, Medical University of Vienna, Austria, and of the Division of Cardiology, Clinic Favoriten, Vienna, Austria, using the high-end scanner Vivid E9 (General Electric Medical System, Milwaukee, WI, USA). Images were acquired, and measurements were taken according to current guidelines [16,17,18].

The presence of pericardial effusion was described without the requirement of a certain minimum width. Functional parameters such as left ventricular ejection fraction (EF; normal ≥55%) were calculated using the biplane Simpson method as well as the left ventricular global longitudinal strain (GLS). Speckle tracking imaging was performed, extending the standard protocol, after image acquisition on a modern offline clinical workstation equipped with dedicated software (EchoPAC; GE Healthcare, Wauwatosa, WI, USA).

### 2.8. Cardiac Magnetic Resonance Imaging

Cardiac MRI was acquired on a Magnetom Avanto 1.5 tesla scanner (Siemens Healthineers, Erlangen, Bayern, Germany) at the Department of Radiology, Clinic Favoriten, Vienna, Austria. A standardized protocol was followed for morphological and functional cardiac evaluation. Cine-sequences were acquired in 2-, 3- and 4-chamber views as well as short axis view. T1 and T2 mapping, both native and with contrast medium (Clariscan, 0.3 mL/kg/body weight, max. 30 mL, GE-Healthcare Oslo, Norway), were performed in the 4-chamber view and short axis view. Extracellular volume (ECV) was calculated from the midseptal region in pre- and postcontrast T1 mapping using a hematocrit value drawn immediately before MRI. Late enhancement was assessed in phase-sensitive inversion recovery (PSIR) sequences in 2-, 3- and 4-chamber views as well as short axis view. The presence of pericardial effusion was described without the requirement of a certain minimum width.

The MRI images were analyzed by both a senior cardiologist and a radiologist.

### 2.9. Chest Computed Tomography

A low-dose chest CT (Somatom Definition Edge, Siemens Healthineers, Erlangen, Bayern, Germany) scan was performed without contrast. The scanning programs used were CareDose4D (Siemens Healthineers, Erlangen, Bayern, Germany) and CarekV (Siemens Healthineers, Erlangen, Bayern, Germany) to optimize the tube voltage. Parameters were used as follows: tube voltage 120 KV, tube current standard (reference mAS 200-50), pitch 1.2, rotation time 0.5 s and collimation 128 × 0.6 mm.

Images were reconstructed in multiplane reformation, axial and coronal. In lung and soft tissue, window images were reconstructed with 1.0 and 3.0 mm slice thickness.

### 2.10. Pulmonary Function Test

In the pulmonary function test (Amedtec ECG pro, Aue-Bad Schlema, Sachsen, Germany), vital capacity (VC), forced expiratory volume in 1 s (FEV1), the Tiffeneau index and the maximal expiratory flow were measured.

### 2.11. Statistical Analysis

Data were tested for consistency and normality. Results from categorical variables are expressed as absolute numbers and percentages, while continuous variables are shown as mean and standard deviations. NT-proBNP and troponin T were log-transformed with base 10. For the comparison between groups, continuous variables were compared using a bootstrap-t with and without the assumption of variance homogeneity based on 4000 Monte Carlo simulations. Pearson’s chi-square and Fisher’s exact test were applied for discrete variables.

For further analysis, participants were allocated to a Long-COVID or an asymptomatic group in order to identify Long-COVID risk factors. Univariate logistic regression models were applied to test for independent risk factors for Long-COVID, dyspnea and fatigue using asymptotic as well as *p*-values based on Monte Carlo simulation.

All reported tests were two-sided, and *p*-values < 0.05 were considered statistically significant. All statistical analyses in this report were performed by use of NCSS (NCSS 2022, NCSS, LLC. Kaysville, UT, USA), STATISTICA 13 (Hill, T. & Lewicki, P. Statistics: Methods and Applications. StatSoft, Tulsa, OK, USA) and IBM SPSS (IBM Corp. Armonk, NY, USA) version 26.

## 3. Results

### 3.1. Study Participants

A description of the study enrollment process is shown in Figure 1. In brief, patients who had been hospitalized between February 2020 and April 2021 were contacted by phone between July 2020 and October 2021. Of the 941 consecutive patients who had been listed in a registry of previously hospitalized COVID-19 patients, 150 agreed to participate. A total of 194 refused to participate mainly due to advanced age with accompanying immobility, insufficient time resources and/or a lack of interest. Because of in- and out-of-hospital deaths after COVID-19 disease or wrong contact information in the registry, 597 patients were lost to follow-up after hospital discharge, and it was, therefore, not possible to recruit them for the study. The mean interval since hospital discharge was 6.1 ± 1.7 months. None of the participants had received COVID-19 vaccination prior to the infection.

### 3.2. Clinical Parameters

Detailed patient characteristics 6 months after discharge are depicted in Table 1 for the entire study population, Long-COVID patients and asymptomatic patients. In brief, 60 patients (40%) were female and their mean age was 53.5 ± 14.5 years. Overweight and a history of arterial hypertension were the most common comorbidities. Of the 150 study participants, 138 had been admitted to a normal ward during the acute COVID-19 disease and 12 to an ICU. Those patients admitted to an intensive care unit developed Long-COVID more often than patients admitted to a normal ward (*p* = 0.039). The mean duration of COVID-19 hospitalization was 10.9 ± 11.6 days. Patients with a longer in-hospital stay developed Long-COVID more frequently (*p* = 0.001). Within the total study population, 40% received oxygen via nasal cannulae, a further 12% were treated with nasal high-flow oxygen and 6% of the total study population had to be intubated.

Regarding laboratory results, cardiac or inflammation markers were not elevated 6 months post COVID-19; however, compared to asymptomatic patients, those with Long-COVID had significantly higher levels of NT-proBNP (*p* = 0.024) and troponin T (*p* = 0.041).

### 3.3. Imaging Parameters

Imaging parameters 6 months post COVID-19 disease are shown in Figure 2 as well as in Table 2, subdivided into a Long-COVID and an asymptomatic group.

Cardiac manifestations on echocardiography included reduced left ventricular function (i.e., left ventricular ejection fraction ≤ 55%), global longitudinal strain and/or diastolic dysfunction, and on MRI, signs of pericarditis and/or myocarditis. Indeed, of the 108 cardiac MRIs performed, 5 (4%) revealed former or persistent inflammatory changes (Figure 2). A total of 18% showed traces of pericardial effusion (mean of 6 mm and a maximum of 14 mm). Overall, 21% of the total study population showed cardiac manifestation, 23% in the Long-COVID and 14% in the asymptomatic group, without a significant difference between the groups (*p* = 0.25, Table 2).

Pulmonary function tests in the total study population (n = 119) showed a reduced vital capacity in 11%, suggesting restrictive lung disease (Figure 2). There was no significant difference between groups (*p* = 0.91, Table 2).

Chest CT scans revealed pneumonic consolidation, lymph node enlargement, bilateral ground glass opacities and/or fibrosis in 24% of the total study population, in 25% of Long-COVID patients and in 20% of asymptomatic patients with no significant difference between groups (*p* = 0.65, Table 2) and no statistical association with exertional dyspnea (*p* = 0.73).

### 3.4. COVID-19-Related Symptoms

Persisting symptoms were reported by 75% of the patients (n = 113) even 6 months after COVID-19 disease, and they, therefore, fulfilled criteria for Long-COVID (Figure 3A). The two most common symptoms, fatigue and exertional dyspnea, accounted for 62%, whereas 25% of those patients suffered from both. Only 13% of the patients with Long-COVID presented symptoms other than fatigue and exertional dyspnea.

All symptoms decreased in frequency over time (Figure 3B).

### 3.5. Long-COVID Risk Factors

Long-COVID risk factors in general were a longer in-hospital stay (*p* = 0.004), admission to an ICU (*p* = 0.001) and higher NT-proBNP levels (*p* = 0.026, Table 3).

Risk factors for persisting exertional dyspnea were the type of ventilation (intubation presenting the highest risk) (OR 5.1 [95% CI 1.13–22.6], *p* = 0.02), a longer in-hospital stay (OR 1.096 [95% CI 1.035–1.161], *p* = 0.002), admission to an intensive care unit (OR 5.63 [95% CI 1.45–21.76], *p* = 0.012) and/or overweight (OR 5.37 [95% CI 1.95–14.83], *p* = 0.001) (Table 4).

With regard to fatigue, a longer in-hospital stay emerged as the only independent risk factor (OR 1.057 [95% CI 1.008–1.107], *p* = 0.021).

### 3.6. Long-COVID Symptoms of Fatigue and Exertional Dyspnea in Relation to Organ Manifestations

Reported exertional dyspnea was associated with impaired pulmonary function (*p* = 0.026), reduced GLS (*p* = 0.003) and/or left ventricular diastolic dysfunction (*p* = 0.04; Table 4). These findings were apparent only in half of the patients presenting with exertional dyspnea. In the other half, no medical causes could be found. Additionally, no underlying causal conditions were found for fatigue.

There was no correlation between structural and/or functional abnormalities and Long-COVID.

## 4. Discussion

The aim of this analysis was to deliver a comprehensive report on the prevalence of symptoms and possible long-term impairments 6 months after hospitalization for COVID-19 disease, focusing on the cardiopulmonary system. There are five major findings.

Firstly, cardiac manifestations were apparent in 21% of the patients and comprised mainly reduced left ventricular function, reduced GLS and/or diastolic dysfunction on echocardiography and signs of pericarditis and/or myocarditis on MRI. However, whether these findings were pre-existing or caused by COVID-19 cannot be stated with certainty since, in these patients, imaging studies were not performed prior to COVID-19.

Cardiac MRI did show residuals of peri- and myocarditis in five cases, which would translate to a higher incidence compared to the normal population. None of the patients with inflammatory changes had previous COVID-19 vaccination. Our data are consistent with a recent systematic literature review on cardiac involvement in MRIs obtained at a median of 63 days after COVID-19 disease showing a higher incidence of myocarditis in up to 37% but similar percentages of pericarditis in 3% of the patients and pericardial effusions in 15% of the patients [19]. Additionally, our data are in line with a report by Roca-Fernández et al. who found that 19% of Long-COVID patients showed cardiac impairment (defined as reduced EF, GLS or high left or right ventricular end-diastolic volume) 6 months, and for 15%, 12 months, after COVID-19 disease [20]. Whereas the incidence is rather low in comparison to other COVID-19 sequelae, absolute numbers are considerable given the high COVID-19 incidence in the general population during this ongoing pandemic.

Traces of pericardial effusion was detected by MRI in 18% of the patients, which is similar to numbers reported in two published studies with 5–20% approximately 2 months after COVID-19 in both nonhospitalized and hospitalized patients [21,22].

Studies suggest that capillary leakage caused by inflammatory processes might be the cause [23]. In general, it is difficult to distinguish physiological from pathological amounts of pericardial fluid. We employed MRI as the tool of choice for the detection of pericardial effusion. Nonetheless, since pre-COVID-19 images had obviously not been obtained, only further follow-up imaging studies would be able to distinguish between progression, no change or even resolution of pericardial effusion and thus support a causal or coincidental relationship with COVID-19 disease.

Secondly, chest CT scans showed postinfectious changes such as pneumonic consolidation, lymph node enlargement, bilateral ground glass opacities and/or fibrosis in approximately one-fifth of all patients, and findings were related to the severity of the COVID-19 disease.

Published data show 30% abnormal CT scans after 6 months, which matches our findings of approximately 20% very well [24]. Restrictive pulmonary disease (11% in our study) and small airway dysfunction have been published already with a comparable incidence of 15% [25,26]. Follow-up CTs will reveal how long they persist or if they are even irreversible.

Thirdly, six months after discharge, the majority of previously hospitalized COVID-19 patients still suffered from at least one symptom, such as chronic fatigue and/or exertional dyspnea.

Reviewing data from published studies [27], the percentage of patients suffering from Long-COVID after hospitalization varies between 40 and 87%, with the two dominating symptoms being exertional dyspnea and chronic fatigue [28,29,30].

This reported prevalence is in agreement with the prevalence of Long-COVID in our study of 75% in previously hospitalized patients, with the same leading symptoms of fatigue and exertional dyspnea. Encouragingly, many patients already experienced an improvement in the reported symptoms, and follow-up will show how long Long-COVID symptoms persist. Additionally, a recent study demonstrated that the incidence of Long-COVID was reduced by 50%, suggesting that increasing rates of vaccination might attenuate or even reduce rates of Long-COVID [31].

Fourthly, the most important independent risk factor for Long-COVID was the severity of the disease course during hospitalization, i.e., a longer in-hospital stay and/or admission to an ICU increased the risk. This is consistent with previous studies [11,12]. Additionally, NT-proBNP levels, even with a normal median (63.9, IQR 30.9–132.3), were also an independent risk factor for the development of Long-COVID. This suggests that those developing Long-COVID show clinical or subclinical cardiac involvement leading to higher cardiac hormones within the COVID-19 disease course. Another assumption could be that patients with previously higher levels, due to underlying diseases, might be more likely to develop Long-COVID. It is difficult to say whether cardiac involvement is the cause or if it merely coincides.

Fifthly, with regards to the persistence of symptoms, in at least half of the patients, exertional dyspnea was associated with echocardiographic abnormalities such as reduced GLS, diastolic dysfunction and/or an impaired pulmonary function due to reduced vital capacity, providing evidence for a cardiac and/or pulmonary cause of symptoms. Interestingly, bilateral ground glass opacities as well as fibrosis secondary to COVID-19 do not necessarily result in exertional dyspnea.

Fatigue is a leading symptom of Long-COVID [32]. In our study, we could not identify any specific clinical manifestation that could have served as a plausible explanation for this symptom. Similarly, the pathomechanisms of the post-viral fatigue syndrome after several other viral infections, e.g., Epstein–Barr virus, Cytomegalovirus, Coxsackieviruses and other Coronaviruses, have not yet been unraveled, and there is hope that COVID-19 research will help shed light on this long-term sequelae in the near future [33].

## 5. Limitations

Firstly, because a relatively high number of registry patients had either been lost to follow-up (n = 597) or refused to participate (n = 194), an inherent bias of the reported results cannot be excluded.

Secondly, a larger sample size would have been favorable, as it is likely that additional predictors might have been found. Furthermore, a meaningful stratification between non-ICU and ICU patients was not possible due to the low number of ICU patients (n = 12).

Moreover, this study did not sequence the various SARS-CoV-2 subtypes, even though, through the time course of the study, different subtypes occurred.

Another shortcoming of the study is that the diagnosis of myocarditis was not confirmed by myocardial biopsy analyses.

## 6. Conclusions

Even 6 months after recovery from COVID-19, the majority of previously hospitalized patients still suffer from at least one symptom, such as chronic fatigue and/or exertional dyspnea.

While there was no association between chronic fatigue and cardiopulmonary abnormalities, impaired pulmonary function, reduced GLS and/or diastolic dysfunction were significantly more prevalent in patients presenting with exertional dyspnea.

Structural and functional abnormalities were less frequent compared to the portrayed symptoms, and it remains a challenge to substantiate the symptoms.

In general, from a cardiac perspective, it can be said that long-term effects are rather rare but exist and that there are hints of previous cardiac involvement.

Follow-up examinations will be needed in order to elucidate which abnormalities are related to COVID-19 or to other comorbidities and whether they are reversible.

## Figures and Tables

**Figure 1 jcm-12-01536-f001:**
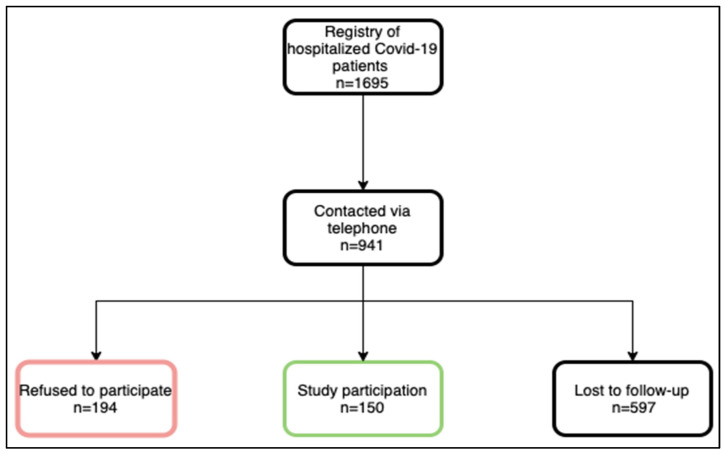
Description of the study enrollment process. n = number of patients.

**Figure 2 jcm-12-01536-f002:**
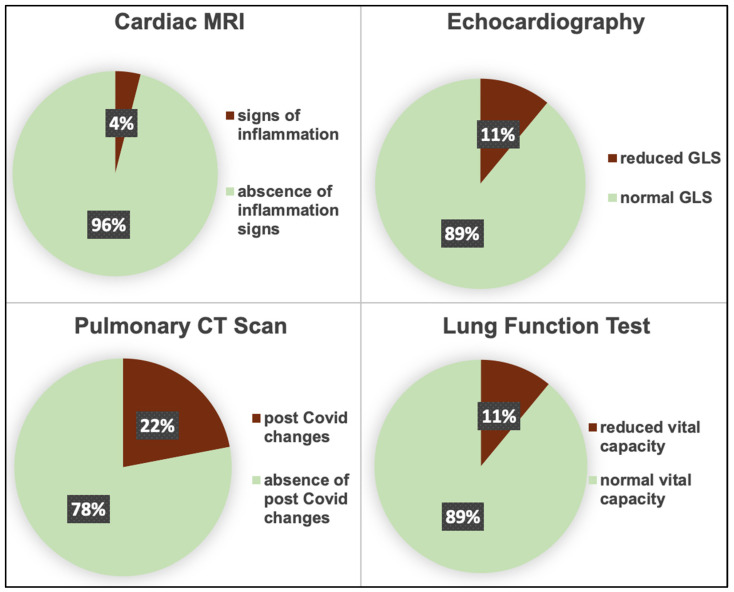
Cardiac and pulmonary structural and functional changes 6 months post COVID-19. MRI, magnetic resonance imaging; GLS, global longitudinal strain; CT, computed tomography.

**Figure 3 jcm-12-01536-f003:**
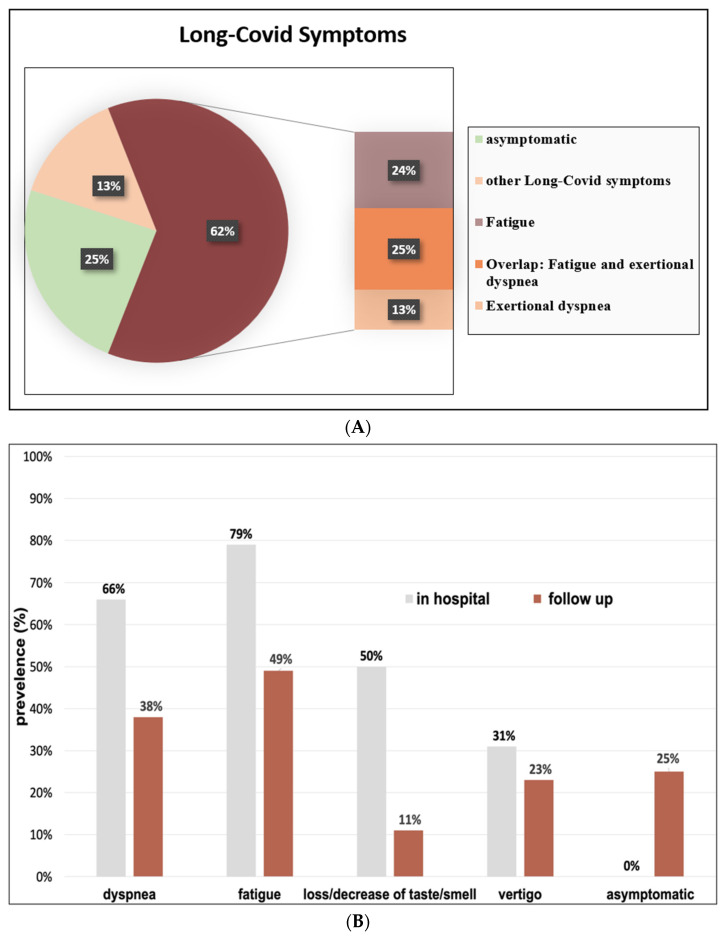
(**A**) Spectrum of Long-COVID symptoms 6 months post discharge. (**B**) Spectrum of Long-COVID symptoms 6 months post discharge in relation to acute phase symptoms during hospital stay.

**Table 1 jcm-12-01536-t001:** Patient characteristics 6 months post discharge and details of in-hospital stay of 150 patients analyzed as total study population and according to symptoms.

Patient Characteristics	Total Study Population (n = 150)	Long-COVID (n = 113)	Asymptomatic at Follow-Up (n = 37)	*p*-Value
Age, years	53.5 ± 14.49	56.48 ± 12.56	53.67 ± 12.80	0.37
Female, %	60.0 (40.0)	48.0 (42.5)	12.0 (32.4)	0.28
BMI, kg/m^2^	29.01 ± 5.44	29.75 ± 4.99	30.16 ± 4.80	0.73
SBP, mmHg	135.55 ± 19.21	137.52 ± 20.91	137.29 ± 19.44	0.47
DBP, mmHg	84.37 ± 10.94	83.97 ±12.27	85.79 ± 10.39	0.99
Heart rate, bpm	70.91 ± 9.81	70.60 ± 9.62	69.58 ± 9.82	0.57
SpO_2_, %	**97.79 ± 1.20**	**97.62 ± 1.29**	**98.25 ± 1.15**	**0.03**
6-MWT, m	556.7 ± 99.30	549.6 ± 97.30	577.7 ± 104.10	0.23
Arterial hypertension, %	68 (45.3)	53 (46.9)	15 (40.5)	0.50
Coronary artery disease, %	11 (7.3)	10 (8.8)	1 (2.7)	0.213
Pre-existing heart failure, %	4 (2.7)	4 (3.5)	0 (0)	0.25
Overweight, %	112 (75.7)	85 (75.9)	27 (75.0)	0.91
Diabetes mellitus, %	29 (19.3)	20 (17.7)	9 (24.3)	0.38
COPD or Asthma, %	16 (10.7)	14 (12.4)	2 (5.4)	0.23
Hemoglobin, g/dL	15.15 ± 11.30	16.57 ± 17.70	14.86 ± 1.22	0.51
eGFR, mL/min/1.73 m^2^	86.56 ± 21.42	88.08 ± 20.76	93.11 ± 17.08	0.25
CRP, mg/dl	0.25 ± 0.39	0.3 ± 0.52	0.27 ± 0.29	0.83
NT-proBNP, pg/mL ^#^	**1.79 ± 0.52**	**1.84 ± 0.51**	**1.62 ± 0.53**	**0.024**
Troponin T, ng/L ^#^	**0.78 ± 0.28**	**0.81 ±0.27**	**0.70 ± 0.27**	**0.041**
CK, U/L	139.1 ± 94.00	160 ± 121.9	132.2 ± 82.10	0.21
Normal ward, %	138 (92.0)	101 (89.4)	37 (100.0)	**<0.00001**
Intensive care ward, %	12 (8.0)	12 (10.6)	0 (0.0)	**0.039**
Length of in-hospital stay	9 (5.0–12.0)	9.0 (5.5–13.5)	7.0 (4.5–9.0)	**0.001**
Ventilation				
No oxygen, %	60 (40.0)	41 (36.3)	19 (51.4)	0.12
Oxygen, %	61 (40.7)	48 (42.5)	13 (35.1)	0.45
Nasal high flow, %	18 (12.0)	13 (11.5)	5 (13.5)	0.77
NIV, %	2 (1.3)	2 (1.8)	0 (0.0)	1.0
Intubation, %	9 (6.0)	9 (8.0)	0 (0.0)	0.11

SBP, systolic blood pressure; DBP, diastolic blood pressure; bpm, beats per minute; SpO_2_, peripheral oxygen saturation; 6-MWT, 6-minute walk test; COPD, chronic obstructive pulmonary disease; eGFR, estimated glomerular filtration rate calculated using the Cockroft Gault formula; CRP, C-reactive protein; NT-proBNP, N-terminal pro brain natriuretic peptide; CK, creatine kinase. Overweight was defined as a BMI of ≥25 kg/m^2^ according to the definition of the World Health Organization. Categorical variables are shown as absolute numbers and percentages, while continuous variables are given as means with standard deviations. *p*-values are based on bootstrap *t*-tests for continuous variables and Pearson’s chi-square or Fisher’s exact for discrete variables. ^#^ log-transformed (base 10).

**Table 2 jcm-12-01536-t002:** Imaging parameters of the 150 study participants 6 months post COVID-19 disease.

Imaging Parameters	Total Study Population (n = 150)	Long-COVID (n = 113)	Asymptomatic at Follow-Up (n = 37)	*p*-Value
Cardiac organ manifestation (n, %)	31 (20.7)	26 (23.4)	5 (13.8)	0.25
Echocardiography				
EF, %	60.69 ± 6.71	60.51 ± 6.81	61.21 ± 6.34	0.60
GLS, %	−18.77 ± 2.93	−18.54 ± 2.78	19.47 ± 3.30	0.21
Diastolic dysfunction, (n, %)	6 (4.0)	5 (4.7)	1 (3.1)	0.64
Cardiac MRI				
EF, %	59.12 ± 8.84	59.11 ± 9.76	59.35 ± 4.78	0.88
LGE, (n, %)	10 (6.7)	3 (12.0)	7 (8.4)	0.69
Myocarditis/pericarditis, (n, %)	5 (4.6)	4 (3.7)	1 (0.9)	0.93
Pericardial effusion, (n, %)	19 (18.0)	5 (20.0)	14 (16.8)	0.71
T1 time, ms	1025.20 ± 108.17	1021.59 ± 121.43	1037.04 ± 41.49	0.38
ECV, %	25.63 ± 3.59	25.59 ± 3.86	25.73 ± 2.56	0.82
Pulmonary organ manifestation (n, %)	34 (23.6)	27 (24.7)	7 (20.0)	0.65
Chest CT				
Post COVID changes, (n, %)	28 (21.5)	23 (23.5)	5 (15.6)	0.35
Pulmonary function test				
Reduced VC (n, %)	13 (11)	3 (11.2)	10 (10.3)	0.91

LVEF, left ventricular ejection fraction; LV-GLS, left ventricular global longitudinal strain measured using speckle tracking imaging; cMRI, cardiac magnetic resonance imaging; LGE, late gadolinium enhancement; ECV, extracellular volume; CT, computed tomography; VC, vital capacity. Categorical variables are shown as absolute numbers and percentages, while continuous variables are given as means with standard deviations. *p*-values are based on bootstrap *t*-tests for continuous variables and Pearson’s chi-square or Fisher’s exact for discrete variables.

**Table 3 jcm-12-01536-t003:** Risk factors for developing Long-COVID.

Risk Factors for Long-COVID	*p*-Value ^#^	Odds Ratio	95% CI
Age	0.07	1.03	0.998–1.05
Gender	0.28	0.65	0.30–1.42
Admission to intensive care unit	**0.001**	- ^c^	- ^c^
Overweight	0.91	1.05	0.44–2.50
Previous illness	0.56	1.27	0.57–2.83
NT-proBNP ^a^	**0.026**	**2.54**	**1.19–5.78**
Troponin T ^a^	0.054	2.04	0.999–4.17
Ventilation			
No oxygen	-	1 ^b^	-
Oxygen	0.20	1.71	0.75–3.88
Nasal high flow	0.75	1.21	0.38–3.89
NIV	1.00	- ^c^	- ^c^
Intubation	1.00	- ^c^	- ^c^
Length of in-hospital stay	**0.004**	**1.14**	**1.05–1.26**
Pericardial effusion in MRI	0.72	0.81	0.26–2.53
Reduced LVF in echocardiography	0.57	1.89	0.21–16.8
Normal GLS in echocardiography	-	1.0 ^b^	-
Borderline GLS in echocardiography	0.25	2.0	0.61–6.59
Reduced GLS in echocardiography	0.37	1.87	0.48–7.2
CT postcovid Changes	0.35	1.66	0.57–4.79
Reduced pulmonary function	0.89	1.10	0.28–4.29

NT-proBNP, N-terminal pro brain natriuretic peptide; MRI, magnetic resonance imaging; LVF, left ventricular ejection fraction; GLS, left ventricular global longitudinal strain; CT, computed tomography. ^#^ *p*-values based on univariate logistic regression; ^a^ log (base 10)-transformed; ^b^ reference category; ^c^ cannot be computed.

**Table 4 jcm-12-01536-t004:** Risk factors for developing exertional dyspnea or fatigue after hospital admission due to COVID-19 disease.

**Risk Factors for Exertional Dyspnea**	** *p* ** **-Value ^#^**	**Odds Ratio**	**95% CI**
Age	0.11	1.02	0.996–1.04
Gender	0.15	0.61	0.31–1.12
Admission to intensive care unit	**0.012**	**5.63**	**1.45–21.8**
Overweight	**0.001**	**5.37**	**1.95–14.8**
Previous illness	0.22	1.61	0.75–3.43
NT-proBNP ^a^	0.16	1.00	1.00–1.00
Troponin T ^a^	0.78	1.08	0.60–1.93
Ventilation	**0.02**	**5.1**	**1.13–22.6**
No oxygen	-	1.0 ^b^	-
Oxygen	0.36	1.43	0.66–3.07
Nasal high flow	**0.014**	**3.98**	**1.32–12.0**
NIV	0.52	2.52	0.15–42.8
Intubation	**0.034**	**5.06**	**1.13–22.6**
Length of in-hospital stay	**0.002**	**1.096**	**1.04–1.16**
Pericardial effusion in MRI	0.87	1.1	0.39–3.1
Reduced LVF in echocardiography	**0.04**	**9.76**	**1.11–86.7**
Normal GLS in echocardiography	-	1 ^b^	-
Borderline GLS in echocardiography	0.33	0.57	0.19–1.74
Reduced GLS in echocardiography	**0.005**	**5.24**	**1.64–16.7**
Diastolic dysfunction	**0.03**	**2.41**	**1.09–5.35**
CT postcovid Changes	0.73	1.16	0.50–2.71
Reduced pulmonary function	**0.028**	**3.62**	**1.15–11.4**
**Risk factors for Fatigue**	** *p* ** **-Value ^#^**	**Odds Ratio**	**95% CI**
Age	0.57	1.01	0.98–1.03
Gender	0.26	0.69	0.36–1.32
Admission to intensive care unit	0.22	2.18	0.63–7.58
Overweight	0.77	1.12	0.53–2.37
Previous illness	0.43	1.34	0.66–2.72
NT-proBNP ^a^	0.55	1.09	0.83–1.43
Troponin T ^a^	0.64	1.14	0.65–2.02
Ventilation			
No oxygen	-	1.0 ^b^	-
Oxygen	0.17	1.65	0.80–3.39
Nasal high flow	0.30	1.75	0.61–5.06
NIV	1.0	- ^c^	- ^c^
Intubation	0.17	2.8	0.64–12.3
Length of in-hospital stay	**0.021**	**1.06**	**1.01–1.11**
Reduced LVF in echocardiography	0.94	0.94	0.18–4.83
Normal GLS in echocardiography	-	1 ^b^	-
Borderline GLS in echocardiography	0.052	2.5	0.96–6.50
Reduced GLS in echocardiography	0.061	2.75	0.91–8.29
Diastolic dysfunction	0.74	1.13	0.55–2.30
CT postcovid Changes	0.395	1.44	0.62–3.35
Reduced pulmonary function	0.91	0.94	0.32–2.796

NT-proBNP, N-terminal pro brain natriuretic peptide; LVF, left ventricular ejection fraction; GLS, left ventricular global longitudinal strain; CT, computed tomography. ^#^ *p*-values based on univariate logistic regression; ^a^ log (base 10)-transformed; ^b^ reference category; ^c^ cannot be computed.

## Data Availability

The data presented in this study are available on request from the corresponding author.

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
