# Peer review of "Cardiopulmonary Long-Term Sequelae in Patients after Severe COVID-19 Disease"

_jcm, 2023, doi:10.3390/jcm12041536_

Round 1
Reviewer 1 Report
After examining the scientific study, the following considerations may be made. The scientific study is well structured in all its parts. In particular, the premises with which the authors introduced the analysis are clear. The objectives that led the authors to carry out this study and the section on materials and methods are equally clear. Particular appreciation can also be expressed for the material on which the study was carried out. The data was collected methodically and without bias. The results were consistent and significant and allowed a discussion section full of food for thought. The authors then developed a discussion of the results achieved.
This article is particularly current as it is particularly relevant to the issue of the covid-19 pandemic.
The number and quality of the citations are appropriate, however, the scientific relevance of the article could benefit from an expansion of the same. In the specific advice to add the following quotes:
· Line 51, I suggest underlining the fact that despite vaccines being introduced many adverse events occurred. Thus I suggest adding the following quote: "Fazio ND, Delogu G, Bertozzi G, Fineschi V, Frati P. SARS-CoV2 Vaccination Adverse Events Trend in Italy: A Retrospective Interpretation of the Last Year (December 2020-September 2021). Vaccines (Basel). 2022 Jan 30;10(2):216. doi: 10.3390/vaccines10020216. PMID: 35214674; PMCID: PMC8880467.
· Line 59, I suggest adding the following quote: “Maiese A, Frati P, Del Duca F, Santoro P, Manetti AC, La Russa R, Di Paolo M, Turillazzi E, Fineschi V. Myocardial Pathology in COVID-19-Associated Cardiac Injury: A Systematic Review. Diagnostics (Basel). 2021 Sep 8;11(9):1647. doi: 10.3390/diagnostics11091647. PMID: 34573988; PMCID: PMC8472043.”
English is well structured in syntax and grammar.
Reviewer 2 Report
“severe Covid-19 infection”, COVID-19 is not an infection
Should abstract methods be expanded, design.. intervention.. outcome? It seems you evaluated the hospitalized patients, but was there a follow up? It is not clear
Conclusions are excellent and not solid, given the low lower limit of the confidence interval, but I would soften them with "it appears, it seems".
53 The Covid-19 infection does not exist. COronaVIrus 19 disease! the infection is from SARS-CoV-2. SARS-CoV-2 infection does not always lead to COVID-19 disease..
54 I suggest a statement like this: <<However, COVID-19 could have detrimental sequelae even after the post-acute phase, representing a new pathological condition: the "post-COVID-19 syndrome (PCS)" or "long-term COVID-19". ” >>
reference: https://www.mdpi.com/2076-3417/12/17/8593
59 6-minute walking distance (6-MWD), I suggest 6-minute walking test (6MWT).. it is more common as an acronym
The methods are better divided by Design and Setting, Population (especially exclusion criteria: age? BMI cut-off? severe heart failure?), Outcome with various subparagraphs that you measured
177 any more details on rejection and loss in follow up?
185 I would have made a stratification between ICU and non-ICU
I would like to know how much a 15% length of hospital stay is linked to the 12 patients in intensive care who generally have a longer period of hospitalization
The limitations at the end of the discussion are missing.. one above all that the odds ratios are affected by a lower CI limit close to 1.
Author Response
Please see the attachment from February 10th, as one point has been changed due to the statisticians response.
Thank you very much- I apologize for uploading 2 versions!

Round 2
Reviewer 1 Report
After examining the scientific study, the following considerations may be made. The scientific study is well structured in all its parts. In particular, the premises with which the authors introduced the analysis are clear. Equally clear are the objectives that led the authors to carry out this study and the section on materials and methods. Particular appreciation can also be expressed for the material on which the study was carried out. The data was collected methodically and without bias. The results were consistent and significant and allowed a discussion section full of food for thought. The authors then developed a discussion of the results achieved.